# Minimally Invasive Spine Stabilization for Pyogenic Spondylodiscitis: A 23-Case Series and Review of Literature

**DOI:** 10.3390/medicina58060754

**Published:** 2022-06-01

**Authors:** Shinichi Ishihara, Haruki Funao, Norihiro Isogai, Masayuki Ishihara, Takanori Saito, Ken Ishii

**Affiliations:** 1Department of Orthopaedic Surgery, Ota Memorial Hospital, Gunma 373-8585, Japan; shintarou198088@hotmail.com; 2Department of Orthopaedic Surgery, School of Medicine, International University of Health and Welfare (IUHW), Narita 286-0048, Japan; n.isogai0813@gmail.com; 3Department of Orthopaedic Surgery, International University of Health and Welfare Mita Hospital, Tokyo 108-8329, Japan; 4Department of Orthopaedic Surgery, International University of Health and Welfare Narita Hospital, Narita 286-8520, Japan; 5Department of Orthopaedic Surgery, Kansai Medical University, Osaka 573-1191, Japan; ishihara0714@yahoo.co.jp (M.I.); saitot@takii.kmu.ac.jp (T.S.)

**Keywords:** minimally invasive spinal treatment (MIST), minimally invasive spine surgery (MISS), minimally invasive spine stabilization (MISt), pyogenic spondylodiscitis, percutaneous pedicle screw (PPS)

## Abstract

*Background and Objectives:* The incidence of pyogenic spondylodiscitis has been increasing due to the aging of the population. Although surgical treatment is performed for refractory pyogenic spondylodiscitis, surgical invasiveness should be considered. Recent minimally invasive spine stabilization (MISt) using percutaneous pedicle screw (PPS) can be a less invasive approach. The purpose of this study was to evaluate surgical results and clinical outcomes after MISt with PPS for pyogenic spondylodiscitis. *Materials and Methods:* Clinical data of patients who underwent MISt with PPS for pyogenic spondylitis were analyzed. *Results:* Twenty-three patients (18 male, 5 female, mean age 67.0 years) were retrospectively enrolled. The mean follow-up period was 15.9 months after surgery. The causative organism was identified in 16 cases (69.6%). A mean number of fixed vertebrae was 4.1, and the estimated blood loss was 145.0 mL. MISt with PPS was successfully performed in 19 of 23 patients (82.6%). Four cases (17.4%) required additional anterior debridement and autologous iliac bone graft placement. CRP levels had become negative at an average of 28.4 days after surgery. There was no major perioperative complication and no screw or rod breakages during follow-up. *Conclusions:* MISt with PPS would be a less invasive approach for pyogenic spondylodiscitis in elderly or immunocompromised patients.

## 1. Introduction

The incidence of pyogenic spondylodiscitis ranges from 0.2 to 2.4 per 100,000 per year [1,2,3,4] and is more common in males [5]. In recent years, the incidence has been on the rise due to the increase in elderly and immunocompromised patients [5,6,7], in addition to advances in imaging techniques such as magnetic resonance imaging (MRI) for diagnosis [8]. Pyogenic spondylodiscitis is often difficult to treat, and mortality rates have been reported to be between 2–20% [5,9,10,11]. One of the reasons for the difficulty in treatment is the low positive rate of the causative organism [12,13,14]. Although some reports recommend early surgery including debridement [15,16], the first-line is conservative treatment [12,17,18]. The most important aspect of conservative treatment is the administration of antibiotics for a sufficient period of time as well as rest and external orthosis [2,5]. However, the failure rate of conservative treatment has been reported to be between 12%and 18% [12,19,20,21,22], and surgical treatment should be considered when there is a refractory infection, progressive bone destruction, excruciating pain, or neurological deterioration.

Traditionally, the ideal treatment is debridement and anterior column reconstruction; however, it is sometimes inadvisable in patients with poor medical conditions. Recent reports have described posterior fixation techniques with an open approach for pyogenic spondylodiscitis [23,24]. Moreover, minimally invasive spine stabilization (MISt) using a percutaneous pedicle screw (PPS) can be a less invasive approach for patients with multiple comorbidities. Although there are a few reports describing the efficacy of PPS [25,26], its superiority is still controversial. The purpose of this study was to evaluate surgical results and clinical outcomes after MISt with PPS for pyogenic spondylodiscitis.

## 2. Materials and Methods

A multicenter retrospective analysis was conducted. This study was approved by the institutional review board at our hospital and all subjects were given an option to opt-out. Patients who underwent MISt with PPS for refractory pyogenic spondylodiscitis in the thoracic and lumbar spine between 2008 and 2014 were enrolled. Medical charts and radiographs were retrospectively reviewed. All patients underwent blood culture, urine culture, sputum culture, or needle biopsy to identify the causative organism. After the culture tests were performed, broad-spectrum antibiotics were administered for a sufficient period of time, and de-escalation was applied when the culture results were available. All patients had adequate conservative treatment before surgery; however, patients presented with residual symptoms, such as intractable pain, neurological deterioration, elevated serum C reactive protein (CRP), and destructive radiographic changes. Basically, posterior instrumentation with 2-level above and 2-level below fixation is considered in MISt with PPS procedure. The fixation range of instrumentation is finally decided after considering the degree of intervertebral disc or vertebral body destruction, age, physical activity, bone strength, and spinal motion segment or junctional lesion. Posterior fixation with PPS was performed, and patients were allowed to move immediately postoperatively, with rehabilitation performed with a corset within 1 week after surgery. If there was a large bony defect or unhealed infection, additional anterior debridement and autologous iliac bone graft placement was performed. 

The following clinical data were collected: age; gender; comorbidities and predisposing condition; level of infection; causative organism; operative time; estimated blood loss; the number of fixed vertebrae; with or without anterior debridement and autologous iliac bone graft placement; time until CRP became negative after surgery; and perioperative complications. Perioperative complications included surgical site infection, postoperative neurologic deficit, postoperative hematoma, rod/screw breakage, implant replacement, and medical complications (symptomatic anemia, pneumonia, urinary tract infection, sepsis, and other) [27]. Major perioperative complications were determined as follows; mortality, massive bleeding, mechanical failure requiring revision, pulmonary embolism, sepsis, and other fatal events. There was no patient who had a history of surgery using artificial implants such as aortic vascular surgery or joint surgery in this study.

## 3. Results

A total of 23 subjects (mean age, 67.0 years old) were enrolled, comprising of 18 men and 5 women. Table 1 summarizes the characteristics of subjects enrolled in the present study. Twenty (87.0%) patients had comorbidities with a history of solid cancer being the most common comorbidity (eight patients, 34.8%). Other comorbidities included diabetes mellitus in five patients (21.7%), renal failure in three patients (13.0%), cerebrovascular disease in three patients (13.0%), liver cirrhosis in two patients (8.7%), angina pectoris in one patient (4.3%), pancreatitis in one patient (4.3%), and depression in one patient (4.3%) (including one duplicate case). The infected intervertebral discs and vertebral bodies were located at the thoracic spine in four patients (17.4%), thoracolumbar in four patients (17.4%), lumbar in eleven patients (47.8%), and lumbosacral in five patients (21.7%) (including one duplicate case) (Table 1). 

Causative organisms for all patients are shown in Table 2. The most common causative organism identified was *Staphylococcus aureus* in six patients (26.1%), followed by methicillin resistant *Staphylococcus aureus* (MRSA) in three patients (13.0%), *Streptococcus dysgalactiae* in two patients (8.7%), *Streptococcus intermedius* in one patient (4.3%), *Streptococcus mutans* in one patient (4.3%), *Escherichia coli* in one patient (4.3%), *Enterobacter aerogens* in one patient (4.3%), and *Corynebacterium* in one patient (4.3%) The causative organism was unknown in seven patients (30.4%) (Table 2). 

A detailed presentation of patient data is shown in Table 3. The mean number of fixed vertebrae was 4.1 (2–6 vertebrae). The mean operative time was 205.1 min (55–399 min), and the mean estimated blood loss was 145.0 mL (5–550 mL).

The mean time until CRP became negative after operation was 28.4 days (10–56 days). Additional anterior debridement and autologous iliac bone graft placement were required in four cases (17.4%). No major perioperative complications were observed, and there were no cases of rod/screw breakage, or implant replacement during follow-up (Table 4).

### Case

50 years old male presented with severe low back pain and fever. Despite the administration of antibiotics for a sufficient period of time, bone destruction progressed and the patient was unable to leave the bed due to pain, thus MISt with PPS was performed (Figure 1 and Figure 2). CRP became negative 22 days postoperatively, and bony union was achieved on CT images at 12 months after surgery (Figure 3). 

## 4. Discussion

In recent years, the incidence of pyogenic spondylodiscitis has been rising due to an increase in the elderly population [5,6,7]. Pyogenic spondylodiscitis is often difficult to treat in elderly patients with multiple comorbidities or immunocompromised patients. It is indisputable that first-line treatment for pyogenic spondylodiscitis is adequate conservative treatment. Multiple randomized controlled trials suggest that antibiotics are required for 6 weeks with or without surgery and 8 weeks or more for *Staphylococcus aureus* and MRSA [28,29,30,31]. Surgical treatment is considered when conservative treatment fails, and the indications for surgical treatment have been reported in the literature: the presence of neurological symptoms, spinal instability, spinal deformity, and failure of conservative treatment [1,2,6,9,10].

In this study, MISt with PPS was successfully performed in 19 of 23 patients (82.6%). Although mortality rates of pyogenic spondylodiscitis have been reported to be between 2% and 20%, all 23 patients were cured of the infection within a mean of 28.4 days without any serious complications. The first reason for this high healing rate was the minimally invasive approach with PPS fixation. The conventional open method leads to massive bleeding, especially in long spinal fixation; however, PPS allows for long spinal fixation without significant bleeding and soft tissue damage. Posterior spinal fixation using PPS is clearly less invasive than the conventional open method, leading to earlier healing and mobilization without fatal complications. The second factor may be that we could administer adequate antibiotics for a sufficient period of time after conducting various culture tests and selecting sensitive antibiotics. The positivity rate of culture results is said to be 42 to 100%, and the high positivity rate of 69.6% in this study was a major advantage in the treatment of pyogenic spondylodiscitis [5,12,13,14]. These factors are thought to have contributed to the high healing rate of refractory pyogenic spondylodiscitis in this series. It is important to detect the causative organisms not only for the treatment of spondylodiscitis but also for the prevention and treatment of surgical site infection with spinal instrumentation that can result in serious consequences.

However, if the infection is not eradicated or bone union is not achieved after MISt with PPS, additional anterior debridement and reconstruction should be considered. In our case series, 4 of 23 (17.4%) patients required additional anterior debridement and autologous iliac bone graft placement; of which, 3 of 4 (75%) patients were presented with spondylodiscitis in the mid to lower lumbar spine. Because there might be hypermobility in the mid to lower lumbar spine, anterior reconstruction was required due to potential instability. Secure minimally invasive sacropelvic fixation is also required to provide a rigid distal foundation for spondylodiscitis in the lumbosacral spine [32]. Though Lin et al. reported that anterior debridement followed by PPS fixation was effective, it might be better to perform anterior debridement when a significant epidural abscess is present [11]. It is also reasonable to perform laminectomy posteriorly with a separate incision for an epidural abscess associated with neurological deficit. Recently introduced techniques such as full endoscopic surgery via a transforaminal approach can also be performed for the debridement of an epidural abscess [25]. The administration of teriparatide may also be effective as an additional treatment to obtain bony union around the infected vertebrae [33]. However, since Morita et al. reported that teriparatide did not suppress the progression of bone destruction in a murine osteomyelitis model, the administration of teriparatide for pyogenic spondylodiscitis remains controversial [34].

The present study has several limitations. The first is that this is a retrospective observational study, and the cohort was small due to the rarity of refractory pyogenic spondylodiscitis. In order to obtain a higher evidence level, it is desirable to conduct a prospective cohort study comparing the conventional open method and MISt with PPS. In addition, a long-term follow-up is required to observe the recurrence of infection, nonunion, mechanical failure, and potential risk of spinal malalignment. Second, PPS using fluoroscopy is not applicable to the cervical and upper thoracic spine, and it is also difficult to perform in osteoporotic and obese patients because the pedicle cannot be clearly visualized under fluoroscopy. Because we excluded the patients with pyogenic spondylodiscitis in the cervical and upper thoracic spine from this study, it can be a limitation to prove the effectiveness of MISt with PPS procedure in those patients.

## 5. Conclusions

In summary, MISt with PPS was successfully performed in 19 of 23 patients (82.6%) with refractory pyogenic spondylodiscitis without serious perioperative complications. MISt with PPS could be a feasible surgical option for refractory pyogenic spondylodiscitis, especially in elderly and immunocompromised patients.

## Figures and Tables

**Figure 1 medicina-58-00754-f001:**
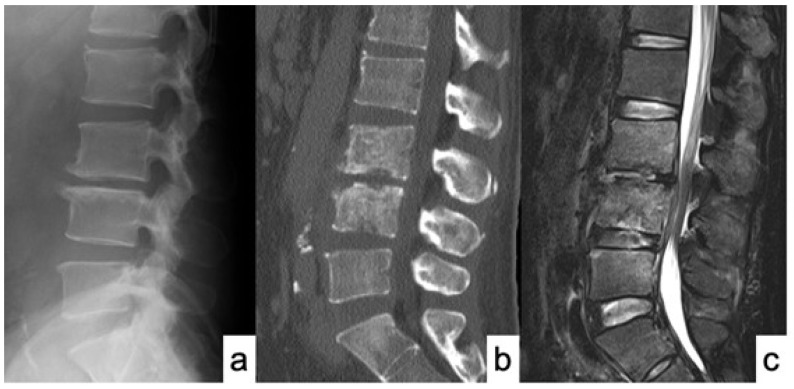
MRI and CT scan of case No. 14 at first admission. (**a**) X-ray of sagittal view; (**b**) Sagittal view of CT scan showed a destructive change of L3 and L4 vertebral bodies; and (**c**) T2-weighted sagittal view of MRI showed spondylodiscitis at L3-4 and L4-5 with epidural abscess.

**Figure 2 medicina-58-00754-f002:**
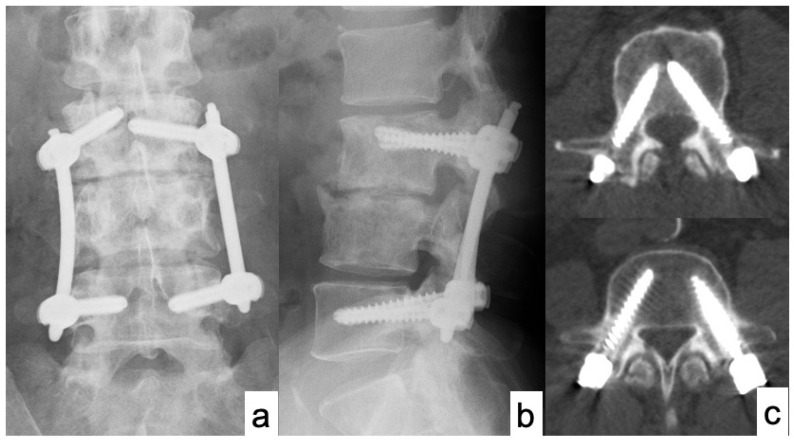
X-ray and CT scan of case No. 14 immediately after surgery. (**a**) X-ray of AP view; (**b**) X-ray of lateral view; and (**c**) Axial views of CT scan showed accurate PPS placements.

**Figure 3 medicina-58-00754-f003:**
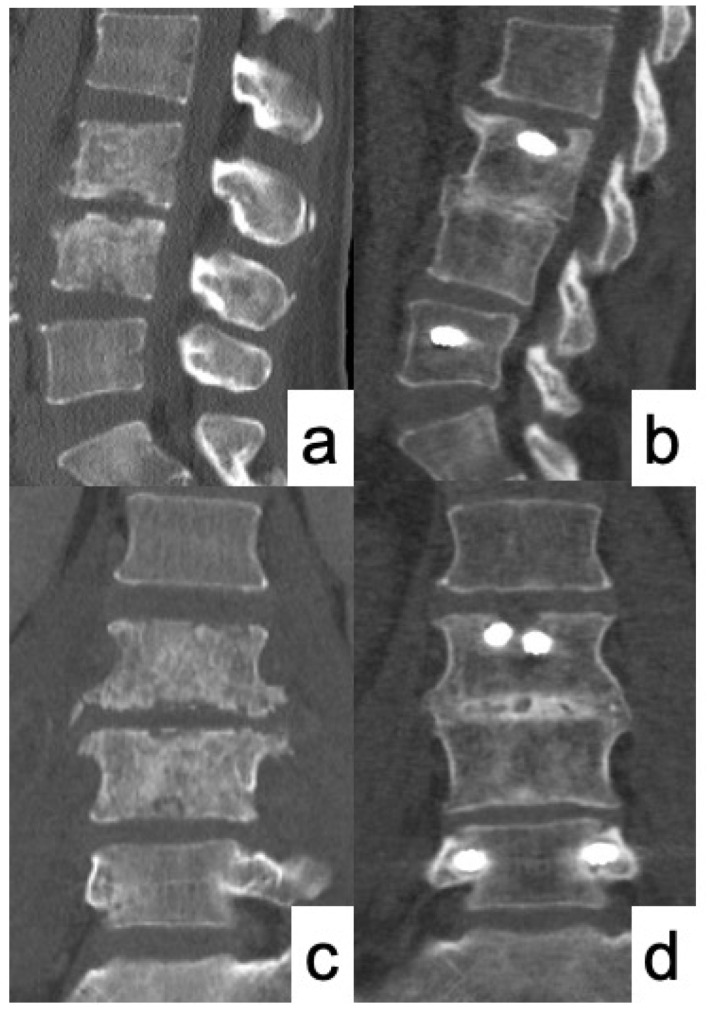
Comparison of CT images before surgery and 12 months after surgery. (**a**,**c**) CT images before surgery showed destructive change of L3 and L4 vertebral bodies; (**b**,**d**) CT image at 12 months after surgery showed good bony union with callus bridging between L3 and L4.

**Table 1 medicina-58-00754-t001:** Demographic data.

Characteristic	*N* = 23 (%)
Gender	
Male	18 (78.3%)
Female	5 (21.7%)
Comorbidities (including duplications)	
Solid cancer	8 (34.8%)
Diabetes mellitus	5 (21.7%)
Renal failure	3 (13.0%)
Cerebrovascular disease	3 (13.0%)
Liver cirrhosis	2 (8.7%)
Angina pectoris	1 (4.3%)
Pancreatitis	1 (4.3%)
Depression	1 (4.3%)
Location (including 1 duplication)	
Thoracic	4 (17.4%)
Thoracolumbar	4 (17.4%)
Lumbar	11 (47.8%)
Lumbosacral	5 (21.7%)

**Table 2 medicina-58-00754-t002:** Causative organisms.

Bacterial Strain	*N* = 23 (%)
*Staphylococcus aureus*	6 (26.1%)
MRSA ^1^	3 (13.0%)
*Streptococcus dysgalactiae*	2 (8.7%)
*Streptococcus intermedius*	1 (4.3%)
*Streptococcus mutans*	1 (4.3%)
*Escherichia coli*	1 (4.3%)
*Enterobacter aerogens*	1 (4.3%)
*Corynebacterium*	1 (4.3%)
Unknown	7 (30.4%)

^1^ MRSA: methicillin resistant *staphylococcus aureus*.

**Table 3 medicina-58-00754-t003:** Detailed presentation of patient data.

Patient No.	Age	Sex	Involved Level	Fixed Vertebrae	Organism	Comorbidities	CRP Become Negative (Days)	Operative Time (min)	EBL (mL)	ADF	F-U (Month)
1	64	M	T7-8	T5-10	Unknown		27	140	50	No	6
2	66	M	T8-9	T6-11	MRSA	Infectious endocarditis	16	122	5	No	6
3	85	F	T9-10	T6-L1	*S. aureus*	Prostate cancer	28	258	70	No	24
4	78	M	T9-10	T7-12	MRSA	Pancreatitis	14	130	300	No	7
5	75	F	T9-11	T6-L2	Unknown		12	278	130	No	6
6	84	M	T10-11	T7-L2	*S. aureus*	DM	28	265	384	No	26
7	69	M	T10-11	T7-L2	Unknown	RF and CD	25	282	67	No	18
8	64	M	L1-2	L1-3	*S. aureus*	Liver Cancer	39	65	20	Yes	6
9	60	M	L2-3	T12-L5	*S. aureus*		11	212	52	No	6
10	62	M	L2-3 L5-S1	T12-S1 (Iliac)	*S. dysgalactiae*	RF	56	364	260	No	6
11	55	M	L3-4	L1-S1	*E. aerogenes*	Colon cancer	47	183	158	Yes	43
12	57	M	L3-4	L1-5	*S. mutans*	Liver cancer	10	245	100	No	16
13	75	M	L3-4	L1-5	*S. dysgalactiae*	DM	51	154	280	No	15
14	50	M	L3-5	L3-5	*S. intermedius*	DM and Liver cancer	56	85	90	No	12
15	39	F	L4-5	L4-5	*S. aureus*	Depression	19	55	20	No	18
16	77	M	L4-5	L2-S1 (S2AI *)	*S. aureus*	Lung cancer	20	193	26	No	6
17	72	M	L4-5	L2-S1 (S2AI *)	Unknown	Gastric cancer	30	181	10	Yes	24
18	57	F	L4-5	L2-S1 (S2AI *)	Unknown		30	215	43	No	18
19	77	M	L4-5	L2-S1 (S2AI *)	Unknown	Lung cancer and angina	30	188	26	No	6
20	77	M	L5-S1	L3-S1 (Illiac)	Unknown	DM	14	265	340	No	13
21	59	M	L5-S1	L3-S1 (Illiac)	MRSA	CD	30	399	550	No	24
22	69	F	L5-S1	L3-S1 (Illiac)	*Coryne* *bacterium*	Uterine cancer	30	261	172	No	36
23	71	M	L5-S1	L3-S1 (S2AI *)	*E. coli*	DM, RF, and CD	30	196	171	Yes	24

* S2AI: S2 alar iliac; MRSA: methicillin resistant *staphylococcus aureus*; DM: diabetes mellitus; RF: renal failure; CD: cerebrovascular disease; ADF: anterior debridement and fusion.

**Table 4 medicina-58-00754-t004:** Perioperative data.

Numbers of fixed vertebrae	4.1 vertebrae (2–6)
Operative time	205.1 min (55–399)
Estimated blood loss	145.0 mL (5–550)
Anterior debridement and bone graft placement	4 cases (17.4%)
CRP becomes negative after surgery	28.4 days (10–56 days)
Major perioperative complication	none

## Data Availability

Not applicable.

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
