# Peer review of "Minimally Invasive Spine Stabilization for Pyogenic Spondylodiscitis: A 23-Case Series and Review of Literature"

_medicina, 2022, doi:10.3390/medicina58060754_

Round 1
Reviewer 1 Report
Thank you for the opportunity to review this article.
This study evaluates the effectiveness of minimally invasive stabilization in supporting the treatment of spondylodiscitis resistant to conservative treatment.
The topic is interesting. However, I have some concerns that need clarification.
1) In my opinion it is necessary to separate the role of surgery into two distinct phases: mechanical stabilization and surgical debridement. Therefore in my opinion the main evidence in this article is that minimally invasive posterior stabilization is effective in contributing to the treatment of spondylodiscitis, even in the absence of surgical debridement. I would like the authors to emphasize this aspect and to discuss it further, also by referring to similar evidence in the literature and possible reasons.
2) Consequently, in my opinion, the 4 patients who underwent anterior debridement as an additional and successive procedure cannot be included in the cohort of patients without distinction. They should be identified as patients in whom minimally invasive posterior treatment without surgical debridement was NOT effective. In fact the Authors stated: "However, if the infection is not eradicated or bone union is not achieved after MISt with PPS, additional anterior debridement and reconstruction should be considered. In our case series, 4 of 23 (17.4%) patients required additional anterior debridement and autologous iliac bone graft placement". What do the authors think?
3) Consequently, the Authors should revise the Conclusions, since it is not true that "MISt with PPS was successfully performed in 23 patients".
4) In the Methods section it is in my opinion necessary to better define how complications were classified (various classifications are available in the literature). Consequently, in the Results section it should be better specified what the Authors mean by the expression "major complications".
5) In the Results section, the description of specific case no. 14 I do not think adds any synthetic information to help interpret the results, so I think it would be more useful to remove it.
Thank you.
Reviewer 2 Report
Q1. How did you decide the timing to undergo the MISt with PPS for these cases? how do you feel about the degree of CRP, fever or pain just before undergoing the surgery?
Q2. I think that the blood culture should be given before the spinal stabilization with spinal instrumentation because of the risk for the surgical site infection. please add the sentence about your consideration for this.
Q3. Did you have any patients in this series who have had a surgery with artificial implant, such like an aortic vascular surgery or a joint surgery.
Q4. How did you treat the case with an unknown organism after surgery. I mean what kind of antibiotics should be used for such a case? I think that this method (MISt with PPS) can make sense if antibiotics could be effective against the causative organisms.
Q5. How do you decide the range of instrumentation for each cases?
Q6. Is the removal of implants necessary? How about the timing?
Q7. I could understand that there was no complication regarding the spinal implants. But do you have any cases who had a recurrent of pyogenic spondylitis or delayed union over F/U period? Because patients in your report have many comorbidities. Please describe about these things clearly. This is a very important to conclude that this surgical intervention is effective for pyogenic spondylitis.
Q8. Short F/U periods is also limitation. Furthermore, the cervical and thoracic spine is not included in this study. These may make it difficult to clarify the efficacy of this method for pyogenic spondylitis.
Round 2
Reviewer 1 Report
The authors have addressed all my concerns and amended the manuscript accordingly. I suggest publication of the article in its current form. Thank you.
Reviewer 2 Report
I appreciated that I could review this manuscript about a new surgical concept for pyogenic spondylitis.
I accept this revised manuscript for the publication.